# Novel Skyrmion and spin wave solutions in superconducting ferromagnets

Shantonu Mukherjee[1, 2, 3, *] and Amitabha Lahiri[1, †]

[1]*S N Bose National Centre for Basic Sciences, Block JD, Sector III, Salt Lake, Kolkata 700106, India*

[2]*Institute of Theoretical Solid State Physics, IFW Dresden, Helmholtzstrae 20, 01069 Dresden, Germany*

[3]*Institute for Solid State and Materials Physics, TU Dresden D-01062 Dresden, Germany*

(Dated: July 2, 2024)

The coexistence of Ferromagnetism and superconductivity in so called ferromagnetic superconductors is an intriguing phenomenon which may lead to novel physical effects as well as applications. Here in this work we have explored the interplay of topological excitations, namely vortices and skyrmions, in ferromagnetic superconductors using a field theoretic description of such systems. In particular, numerical solutions for the continuous spin field compatible to a given vortex profile are determined in absence and presence of a Dzyaloshinskii-Moriya interaction (DMI) term. The solutions show that the spin configuration is like a skyrmion but intertwined with the vortex structure — the radius of the the skyrmion-like solution depends on the penetration depth and also the polarity of the skyrmion depends on the sign of the winding number. Thus our solution describes a novel topological structure- namely a skyrmion-vortex composite. We have also determined the spin wave solutions in such systems in presence and absence of a vortex. In absence of vortex frequency and wave vector satisfy a cubic equation which leads to various interesting features. In particular, we have shown that in the low frequency regime the minimum in dispersion relation shifts from $k = 0$ to a non zero $k$ value depending on the parameters. We also discuss the nature of spin wave dispersion in the $\omega \sim \tilde{m}$ regime which shows a similar pattern in the dispersion curve. The group velocity of the spin wave would change it's sign across such a minimum which is unique to FMSC. Also, the spin wave modes around the local minimum looks like roton mode in superfluid and hence called a magnetic roton. In presence of a vortex, the spin wave amplitude is shown to vary spatially such that the profile looks like that of a Néel Skyrmion. Possible experimental signature of both solutions are also discussed.

## I. INTRODUCTION

Superconductivity and ferromagnetism are among the most important phenomena encountered in condensed matter physics. A priori, the properties of superconductors and ferromagnets seem to be totally incompatible with each other. The diamagnetic property of a superconductor prevents magnetic fields from penetrating into the bulk; on the other hand in a ferromagnet, local spins arrange themselves in a certain orientation producing a net magnetization which in turn can produce a magnetic field of their own. Thus it can be expected that controlled interaction among these phenomena may lead to new physics, as well as opening new ways for technological applications. One of the possible ways to explore such an interaction is by forming a heterostructure consisting of ferromagnetic and superconducting layers coupled via an electromagnetic coupling [1] or/and by means of a proximity coupling [2, 3], which is expressed by the Lifshitz invariant [4]. Hybrids of superconductors and ferromagnets have received much attention in the recent past due to their importance in exploring their fundamental aspects, as well as in applications attempting a realization of Majorana fermions [5–8], in superconductor spintronics [9–11], etc. A number of papers in this direction[12–16] have considered the appearance of skyrmions and vortices in ferromagnetic and superconducting layers respectively and studied their interaction. In particular, the possibility of formation of a composite of these topological excitations was studied [1, 3, 12, 15]. Composite objects formed out of vortices and skyrmion are fundamentally interesting and may lead to intriguing applications.

The other way of exploring this interplay of ferromagnetism and superconductivity is to find systems which have a phase where these properties coexist. Such systems are called ferromagnetic superconductors – these are very difficult to find and stabilize in the mentioned phase. One of the possible ways to realize this phase is to use spin triplet superconductivity where the electrons in Cooper pairs form a spin-1 pair. The electrons which give rise to ferromagnetism also form spin triplet Cooper pairs, which then superconduct. Experimental evidence for spin-triplet superconductivity was discovered by Ishida et. al. [17] and by Khaire et. al. [18]. However, there are materials containing periodic lattices of rare earth ions in which, as the temperature is lowered, a superconducting phase is followed by a ferromagnetic phase [19–22]. In such systems the superconducting carriers are not responsible for the

* shantanumukherjeephy@gmail.com

† amitabha@bose.res.in

ferromagnetic transition, rather, the magnetic ions contribute to this ferromagnetic phase. A number of works on such systems were done by Varma and collaborators, who considered the following free energy functional [23, 24]

$$F[\phi, \boldsymbol{A}, \boldsymbol{M}] = \int d^3x \left[ \frac{\rho_s}{2} \left| \left( \boldsymbol{\nabla} - ir_0\boldsymbol{A} \right) \phi \right|^2 + \frac{1}{2}a|\phi|^2 + \frac{1}{4}b|\phi|^4 \right.$$
$$\left. + \frac{1}{8\pi}\boldsymbol{B}^2 + \frac{\rho}{2}|\boldsymbol{\nabla}M^i|^2 + \boldsymbol{B} \cdot \boldsymbol{M} + \frac{1}{2}\alpha|\boldsymbol{M}|^2 + \frac{1}{2}\beta|\boldsymbol{M}|^4 \right]. \tag{1}$$

Here $\phi$ denotes the superconducting order parameter, $\boldsymbol{M}$ denotes the local magnetization density and $\boldsymbol{A}$ is the magnetic vector potential, and the interaction between superconducting and ferromagnetic order is via the Zeeman term ($\boldsymbol{B} \cdot \boldsymbol{M}$). They have argued that the interplay between superconductivity and ferromagnetism can give rise two different phases – one of the phases is described by a transverse oscillatory magnetization with superconductivity preserved and is called spiral magnetic phase for a circular polarization. Another phase is the spontaneous vortex phase (SVP) in which vortices are generated self-consistently by the underlying magnetization. In this work we are interested in the second kind of system for studying the interplay of superconductivity and magnetism which may give rise to a composite topological excitation as well as spin wave excitation having new interesting behaviour. Specifically, we are intrigued by the possibility of emergence of a novel topological arrangement, resulting from the interaction between superconductivity and magnetism expressed by a similar field theoretic framework. This configuration may manifest as a fusion of vortex and skyrmion. The interaction of vortices and skyrmions and the resulting composite formation was previously studied in superconductor-ferromagnet heterostructures as mentioned above. However, the nature of the composite topological configuration has not been studied from a continuum field theoretic perspective as far as we know.

It was known for a long time that the skyrmion solution appears as a finite energy topological configuration in a nonlinear sigma model (in presence of a Dzyaloshinski-Moriya interaction (DMI) term the magnetic skyrmions with a non-zero winding appear as the lowest energy topological configuration or ground state configuration [25]). In our field theoretic model we shall couple such a nonlinear sigma model with a gauged Landau-Ginzburg type theory via an electromagnetic coupling term, more specifically a Zeeman term. As we shall see from the Euler-Lagrange equation, this will lead us to a new coupled structure in which the vortex is coupled to a skyrmion-like configuration. In particular, the solution for the spin field describes a skyrmion-like configuration within the region of extension of the magnetic field due to the vortex configuration. This becomes clearer from the fact that the skyrmion radius (defined below in Sec. II) changes with a change in the vortex length scale, which is the penetration depth of the superconducting component. Outside that region the spin field oscillates with radial distance, which can be compared to the spiral magnetic phase described by Varma et al [23]. Also, we shall see that the solution depends on the sign of the winding number of the vortex. In particular, the solution is skyrmion-like only if the central spin is oriented along the direction of vorticity given by the sign of winding number $N$ of the vortex. This is very interesting as it may have observable consequences. We shall also show that similar results arise in presence of DMI term.

Another interesting aspect of these systems is the behaviour of the spin waves in the bulk. The existence of ferromagnetic order as well as the domain structure in a superconductor are hidden due to the Meissner effect. Due to this, the usual methods of studying magnetic properties of a system like Knight shift or muon spin relaxation are not useful. However, spin waves provide a direct probe into the nature of bulk magnetization, its direction and degree of uniformity. This provides a strong motivation for studying spin waves in such systems. The first study along this direction was done by Ng and Varma[24] and later by Braude and Sonin [26]. They have shown that, unlike other magnetic systems, in such ferromagnetic superconductors the circularly polarized spin waves have two modes having both positive and negative group velocity. In our work we shall show that the frequency $\omega$ satisfy a cubic equation in general. To get a physical picture we shall try to solve the cubic equation with various approximations. We shall show that in the low frequency regime the dispersion relation derived in [26] appears. Also, we shall show that spin wave dispersion curve have a number of local minimum. The excitation around this minimum are similar to roton modes.

Finally, we have solved the spin wave equation in presence of a static vortex along $z$ direction and have shown that in this case, for a given $\omega$ and $k$, the spin wave amplitude depends on the distance from the vortex core.

The plan of the paper is as follows. In Sec II we introduce the model under consideration and through the solutions of Euler-Lagrange equation we try to argue about the coupled skyrmion-vortex structure. In Sec III we introduce the DMI term and discuss how the solution of the spin field changes in presence of this term. In Sec IV we derive the spin wave dispersion relation for different frequency regimes and discuss interesting features of the dispersion curve. Finally, in Sec V we conclude with some discussion on the obtained results and direction for future research.

## II.   NONLINEAR SIGMA MODEL COUPLED TO GINZBURG-LANDAU MODEL

We start by discussing soliton solutions in the nonlinear sigma model [27] which can be expressed by the following Lagrangian,

$$\mathcal{L}_\sigma = \frac{\rho_M}{2} \left( \boldsymbol{\nabla} n^i \right)^2, \tag{2}$$

where $\rho_M$ is called the ground state spin stiffness and is defined as $\rho_M = JS^2 a^{2-d}$. Here $J$ is the coefficient of the exchange coupling term of the microscopic lattice model, $a$ is the lattice spacing and $S$ is spin per site[28]. $\hat{n}$ is the unit vector along the magnetization density $\boldsymbol{M}$ whose magnitude $M_0 = Sa^{-d}$ is assumed to be a constant in this model. This model can be thought of as the continuum version of the Heisenberg model which describes a magnetic system where the amplitude of the magnetic moments is fixed while the direction is position dependent. To derive the Skyrmion solution from this model we write the direction field as the unit vector

$$n^x = \sin F(\rho) \cos \phi,\ n^y = \sin F(\rho) \sin \phi,\ n^z = \cos F(\rho), \tag{3}$$

where $(\rho, \phi, z)$ are cylindrical coordinates in space. We proceed by rewriting the Lagrangian using the ansatz of Eq. (3) so that the Lagrangian becomes

$$\mathcal{L} = \frac{\rho_M}{2} \left( \partial_i F \partial_i F + \frac{1}{\rho^2} \sin^2 F \right). \tag{4}$$

Now we can derive the Euler-Lagrange equation for the field $F(\rho)$ from the above Lagrangian and find the equation

$$\partial_\rho^2 F + \frac{1}{\rho} \partial_\rho F - \frac{\sin 2F}{2\rho^2} = 0. \tag{5}$$

The solution of this equation corresponding to the initial condition that $F(\rho = 0) = \pi,\ \partial_\rho F(\rho = 0) = 1$ behaves as shown in Fig. 1 .

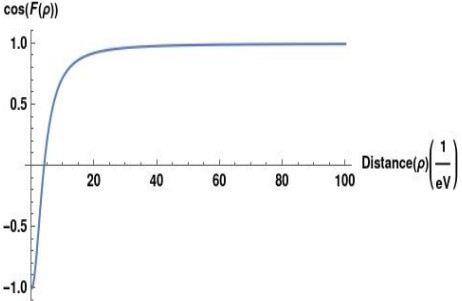

FIG. 1. Skyrmion solution

We note at this point that for different initial value of $\partial_\rho F$ there exist different solution which will differ by skyrmion size (skyrmion radius) [29]. Skyrmions of this type are known as Néel type skyrmions and their polarity is defined as [30]

$$p = \frac{n_z(r = \infty) - n_z(r = 0)}{2}. \tag{6}$$

Thus the polarity of this skyrmion solution can be $p = +1, 0, -1$. We shall now couple this nonlinear sigma model with the Landau-Ginzburg model of superconductor via a Zeeman term which would describe a ferromagnetic superconductor.

The model is described by the Lagrangian

$$\mathcal{L} = -\frac{1}{4} F_{ij}^2 - \frac{\rho_M}{2} \left( \boldsymbol{\nabla} n^i \right)^2 - \frac{\rho_s}{2m} (\boldsymbol{\nabla}\theta - q\boldsymbol{A})^2 - \mu n^i \varepsilon^{ijk} \partial^j A^k. \tag{7}$$

Here $\rho_s$ represent the superfluid density, $q$ and $m$ are the charge and mass of the Cooper pairs and $\mu$ represent the coupling constant of the Zeeman term. This Lagrangian can be compared with the free energy of Eq. (1). Here we

have taken the limit in which the amplitude of the superconducting order parameter is kept fixed and only the phase $\theta$ varies in space. A winding in the phase variable introduces topological vortices into the theory. We have also fixed the amplitude $M_0$ of the magnetization $\boldsymbol{M} = M_0 \hat{n}$. The unit magnetization vector $\hat{n}$ is assumed to vary over a unit sphere in spin space and therefore described by Eq. (3). Putting this ansatz into the Lagrangian (7) we derive the Euler-Lagrange equation for the fields $F(\rho)$ and $\boldsymbol{A}$. These are given by

$$\nabla^2 F - \frac{\sin 2F}{2\rho^2} - \frac{\mu}{\rho_M} \left( \sin F B^z - \cos F \cos \phi B^x - \cos F \sin \phi B^y \right) = 0 \,, \tag{8}$$

$$\left( -\nabla^2 + \frac{q^2 \rho_s}{m} \right) \mathbf{B} = \frac{q\rho_s}{m} \boldsymbol{\Sigma} - \mu \nabla^2 \hat{n} \,. \tag{9}$$

Here we have defined the vorticity $\Sigma^i = \varepsilon^{ijk} \partial^j \partial^k \theta_s$, where $\theta_s$ is the part of the phase which is multivalued and thus creates vortices in the system. As suggested by Eq. (9), in the absence of superconductivity ($\rho_s = 0$) the magnetic field is generated by spin configuration present in the system. In the presence of superconductivity this magnetic field may nucleate vortices in the system. Vortex nucleation by spin textures such as Skyrmions was studied previously by Baumard et al [2] and it was also argued by Greenside et al [23] and by Ng and Varma [24] that in the SVP phase, vortices are spontaneously produced by magnetization. Here we consider that a vortex is created by the spin configuration and try to see what type of spin configuration is compatible to a vortex configuration. For that we first try to solve Eq. (9) using the method of Green functions.

The solution for $\mathbf{B}$ can be written as

$$B^i(x) = \int d^3 x' \, G(x - x') \left[ \frac{q\rho_s}{m} \Sigma^i - \mu \nabla^2 n^i \right] (x'), \tag{10}$$

where $G(x - x')$ is the Green function satisfying the equation

$$\left( -\nabla^2 + \frac{q^2 \rho_s}{m} \right) G(x - x') = \delta^3(x - x'). \tag{11}$$

We can put this solution of Eq.(10) into Eq. (8) to get a nonlinear integro-differential equation for $F$ (or $\hat{\boldsymbol{n}}$) in terms of the sources $\boldsymbol{\Sigma}$ and $\hat{\boldsymbol{n}}$.

$$\begin{aligned}
&\nabla^2 F - \frac{\sin 2F}{2\rho^2} - \frac{\mu}{\rho_M} \sin F \int d^3 x' \, G(x - x') \left[ \frac{q\rho_s}{m} \Sigma^z - \mu \nabla^2 n^z \right] (x') \\
&- \frac{\mu}{\rho_M} \cos F \cos \phi \int d^3 x' \, G(x - x') \left[ \frac{q\rho_s}{m} \Sigma^x - \mu \nabla^2 n^x \right] (x') \\
&- \frac{\mu}{\rho_M} \cos F \sin \phi \int d^3 x' \, G(x - x') \left[ \frac{q\rho_s}{m} \Sigma^y - \mu \nabla^2 n^y \right] (x') = 0,
\end{aligned} \tag{12}$$

In principle one can solve that equation self-consistently to know the resulting spin configuration as well as the vortex configuration which will be intertwined. However, the equation is a nonlinear integro-differential equation which is difficult to solve in general. To get a physical picture out of this equation one may choose a simple vortex configuration and solve for the spin configuration compatible with this, or one can solve for the superfluid velocity $\mathbf{v}_s = \boldsymbol{\nabla} \theta_s$ for a given spin texture like skyrmion. In this work we shall limit ourselves to the first case and shall leave the second possibility for a later work.

Now, to proceed we shall take a straight vortex with the vortex line or string aligned along the $z$ axis. This configuration is expressed by the following equation

$$\boldsymbol{\Sigma} = 2\pi N \int dz' \hat{z} \, \delta^3(\boldsymbol{x} - z'\hat{z}). \tag{13}$$

This ansatz for the vortex configuration will simplify the Eq. (12) by some degree. However, to evaluate the terms like $\sin F \int d^3 x' \, G(x - x') \nabla^2 n^z(x')$ one need to know the function $F(\rho)$ at all points of space. The only possible way to proceed is to solve the equation in an iterative manner. For that we shall ignore such terms in Eq. (12) and shall try to solve the rest part. With all these in mind let us write down the approximated equation

$$\nabla^2 F - \frac{\sin 2F}{2\rho^2} - \frac{\mu}{q\rho_M} \tilde{m}^2 \sin F \int d^3 x' \, G(x - x') \Sigma^z(x') = 0, \tag{14}$$

where we have written $\frac{q\rho_s}{m} = \tilde{m}^2$ which is the photon mass in this case. Writing the superfluid density in natural units as $\rho_s = \frac{m}{2\pi q^2 \lambda^2}$, where $\lambda$ is the London penetration depth, we rewrite the above equation as

$$\nabla^2 F - \frac{\sin 2F}{2\rho^2} - \frac{\mu}{2\pi q\rho_M \lambda^2} \sin F \int d^3x' \; G(x-x')\Sigma^z(x') = 0, \tag{15}$$

Recognizing the Green function $G(x-x')$ for the differential operator $\left(-\nabla^2 + \frac{q^2\rho_s}{m}\right)$ in three spatial dimension to be the Yukawa potential one can perform the integration over the Green function as follows

$$\int dz' \; G(\boldsymbol{x} - \hat{z}\, z') = \int dz' \frac{\exp\left(-\tilde{m}\sqrt{\rho^2 + (z-z')^2}\right)}{\sqrt{\rho^2 + (z-z')^2}} = \int_{-\infty}^{\infty} dx \; e^{-\tilde{m}\rho\cosh(x)} = 2K_0\left(\tilde{m}\rho\right) \tag{16}$$

Finally we get the following simplified equation

$$\partial_\rho^2 F + \frac{1}{\rho}\partial_\rho F - \frac{\sin 2F}{2\rho^2} - N\frac{2\mu}{q\rho_M \lambda^2} \sin F K_0\left(\frac{\rho}{\sqrt{2\pi}\lambda}\right) = 0. \tag{17}$$

We shall solve this equation numerically below. It can be inferred by looking at the equation[31] above that the last term is the contribution from the vortex and may change the solution non trivially. However, for large $\lambda(\lambda \to \infty)$ the last term will decouple from the rest and thus for extremely weak superconductor there will be no coupling between vortex and spin degrees of freedom. The same is also valid for very small penetration depth as the term start to fall rapidly. This can be understood from the plot of the function $\xi = \frac{1}{\lambda^2} K_0\left(\frac{\rho}{\sqrt{2\pi}\lambda}\right)$ against $\lambda$ as we can see from Fig 2.

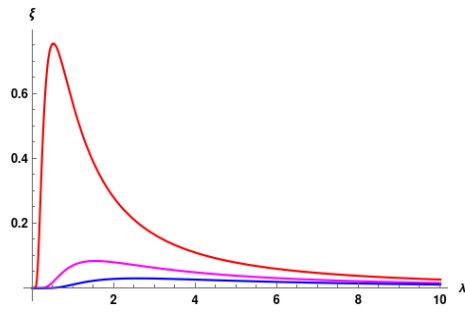

FIG. 2. The function $\xi$ for three different values of distance $\rho$: $\rho=2$ (Red), $\rho=6$ (Magenta), $\rho=10$ (Blue)

It can be shown that in these extreme limits of penetration depth the last term has practically no effect on skyrmion solution. However, for $\lambda \sim 1\text{eV}^{-1}$ i.e. a few $\mu m$ and for $\frac{2\mu}{q\rho_M} \sim 100$ a skyrmion like solution exist only within a region where magnetic field due to the vortex is non zero. This is expressed by the plot of the solution as shown in Fig 3 .

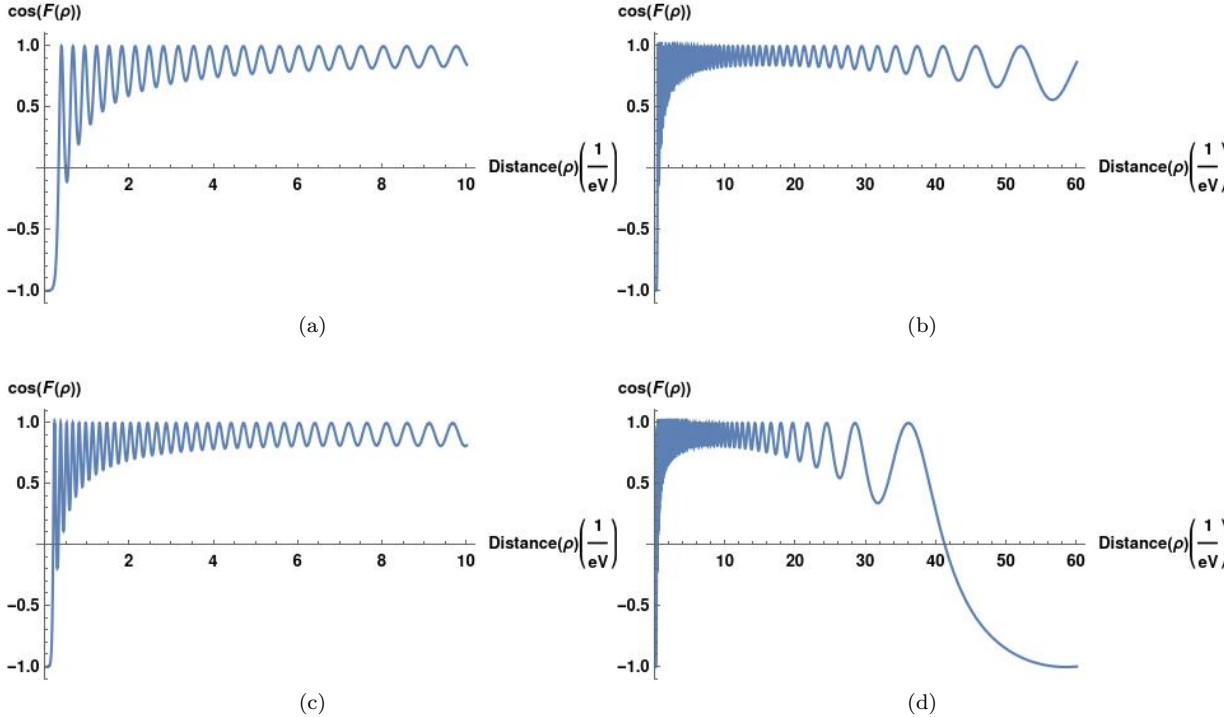

FIG. 3. Skyrmion solution for two different penetration depths is shown. The upper two figures $3a$ and $3b$ are showing the skyrmion like solution for $\lambda = 4$ eV$^{-1}$. The figures $3c,3d$ are showing skyrmion solution for $\lambda = 2$ eV$^{-1}$

We can see from the Fig 3 that with increment in $\lambda$ the distance, over which skyrmion like solution extends, increases. One can also try to measure the changes in the skyrmion radius due to change in $\lambda$. The skyrmion radius$(R_{sk})$ is defined as a distance at which the sign of the $n_z$ changes. We have measured this distance for different values of $1/\sqrt{2\pi}\lambda$. The variation of $R_{sk}$ with penetration depth $\lambda$ is shown in Fig 4 .

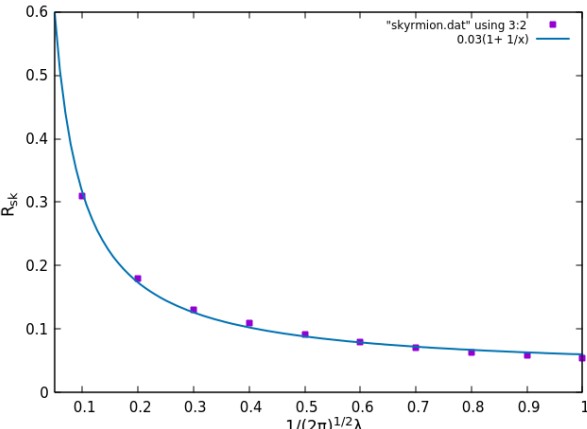

FIG. 4. Skyrmion radius Vs Penetration depth

This shows that skyrmion radius is indeed affected by variation of penetration depth which in turn leads us to the fact that in this solution vortex structure is coupled to skyrmion-like spin configuration. Also, the solution shown above in Fig. 3 is only possible for a specific choice of vorticity $N$. For $N > 0(N < 0)$ the skyrmion like solution exist only if we take the initial condition $n_z = +1(n_z = -1)$ at the origin. The reason for choosing this initial condition is that in the vortex core the magnetic field is very strong and would lead to maximal alignment of the spin field at this point. This shows another proof of coupling of the two topological structures. At this point we must mention that for any choice other than that leads to skyrmion solution at much greater length scale than penetration depth and

do not show any oscillating feature. This shows that for the other choices of the initial condition the two structures decouple.

Thus we see that the skyrmion-like configuration has appeared around the vortex configuration and extends only up to a finite region which is again related to penetration depth of the superconductor. To understand the oscillating nature of the solution we note that in this region a finite magnetic field is present. Thus one first need to understand the nature of skyrmion solution in presence of a constant magnetic field. For that we take a look at the Eq. (8) and let us assume for a moment that the magnetic field is external and have components $\boldsymbol{B} = (0, 0, B)$. For that we get the following equation

$$\nabla^2 F - \frac{\sin 2F}{2\rho^2} - \frac{\mu B}{\rho_M} \sin F = 0. \tag{18}$$

We solve the above equation numerically for a magnetic field $B = -1 \text{ eV}^{-1}$ and find the following solution shown in Fig 5

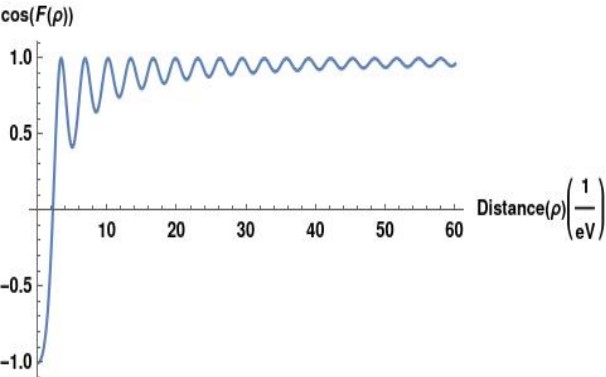

FIG. 5. Skyrmion solution in a constant magnetic field

We see that the solution in this case looks exactly similar to the case of a vortex. The only difference is that, in the case of a constant magnetic field, far from the centre the fluctuations becomes small and curve converges to $n_z = +1$ line. In comparison, in the case of a vortex, the fluctuation about an average continues up to some length scale related to penetration depth and shows wave like oscillation outside that. This can be related to the fact that magnetic field in the system is penetrating only through the vortex and extends only in a finite region. On the other hand the wave like oscillating features outside this region can be related to spiral magnetic phase discussed by Ng and Varma [23].

## III.   DZYALOSHINSKII-MORIYA INTERACTION AND BLOCH SKYRMION

In the previous section we have shown how a coupled structure of vortex and skyrmion-like spin configuration arises in a nonlinear sigma model coupled to Ginzberg-Landau theory of superconductivity. The skyrmion configuration that we considered previously is the Neél type in which spins rotate in radial planes from core to the periphery. However, this configuration is one of the many possible configurations of skyrmions which can be taken into account by a more general ansatz of the form

$$n^z = \cos F(\rho), \ n^x = \sin F(\rho) \cos \Phi(\phi), \ n^y = \sin F(\rho) \sin \Phi(\phi). \tag{19}$$

Here, the field $\Phi$ must be determined from the Euler-Lagrange equation. The general solution for the field $\Phi$ in absence of DMI term is of the form $\Phi = m\phi + \gamma$, where $m$ is the winding number for skyrmion and the phase $\gamma$ decides the helicity. As we shall see, in presence of Dzyaloshinski-Moriya interaction (DMI) the solution takes the specific form $\Phi = \phi \pm \frac{\pi}{2}$. Furthermore, this particular configuration becomes the lowest energy (or ground state) configuration in presence of the DMI term. For this reason DMI term is said to stabilize the skyrmion configuration[25]. We shall try to discuss these points in the context of a nonlinear sigma model in presence of a DMI term which is expressed by the Lagrangian

$$\mathcal{L} = -\frac{\rho_M}{2} \left( \nabla n^i \right)^2 - D\varepsilon^{ijk} n^i \partial^j n^k. \tag{20}$$

With the above generalized ansatz the Lagrangian Eq. (20) becomes

$$\mathcal{L} = -\frac{\rho_M}{2}\left(\partial_i F \partial_i F + \frac{1}{\rho^2}\left(\frac{\partial \Phi}{\partial \phi}\right)^2 \sin^2 F\right) - D\left(\frac{\partial F}{\partial \rho} + \frac{1}{2\rho}\frac{\partial \Phi}{\partial \phi}\sin 2F\right)\sin(\Phi - \phi), \tag{21}$$

The Euler-Lagrange equation for the fields $F$ and $\Phi(\phi)$ are

$$\nabla^2 F - \frac{1}{2\rho^2}\left(\frac{\partial \Phi}{\partial \phi}\right)^2 \sin 2F + \frac{D}{\rho_M \rho}\left(\frac{\partial \Phi}{\partial \phi}\right)\sin(\Phi - \phi)(\cos 2F - 1) = 0, \tag{22}$$

and

$$\frac{1}{\rho^2}\frac{\partial^2 \Phi}{\partial \phi^2} - \frac{D}{2\rho\rho_M}\sin 2F \cos(\Phi - \phi) - \frac{D}{\rho_M}\frac{\partial F}{\partial \rho}\cos(\Phi - \phi) = 0. \tag{23}$$

One can check that the Eq. (23) is trivially satisfied if we take $\Phi = \phi \pm \frac{\pi}{2}$. This choice also leads to minimum energy skyrmion configuration as shown by Nagaosa and Tokura [25] and this configuration is called a Bloch type skyrmion. Putting this solution for $\Phi$ into Eq. (22) we get

$$\nabla^2 F - \frac{1}{2\rho^2}\sin 2F + \frac{D}{\rho_M \rho}(\cos 2F - 1) = 0. \tag{24}$$

As the explicitly $\phi$ dependent terms are absent in the Eq (24) one can now consider cyllindrical symmetry in which $F = F(\rho)$. We note that, this equation leads to skyrmion like configuration only in presence of a potential term in the action. Such a potential term is given below[29]

$$V(F) = K[1 - M_0^2\,(\hat{r}\cdot\hat{n})^2] + \mu M_0\left[H - \hat{n}\cdot\boldsymbol{H}\right], \tag{25}$$

where the first term describes uniaxial magnetocrystalline anisotropy with the coefficient $K$ and $\hat{r}$ being the unit vector in the direction of high symmetry axis [29] ; whereas the second term is the Zeeman energy. In presence of this term the Eq. (24) will be modified as

$$\nabla^2 F - \frac{1}{2\rho^2}\sin 2F + \frac{D}{\rho_M \rho}(\cos 2F - 1) - K \sin F \cos F - \frac{\mu M_0 H}{\rho_M}\sin F = 0. \tag{26}$$

Thus the potential term becomes essential for skyrmion like solution. In the case of a nonlinear sigma model coupled to Ginzberg-Landau model of superconductivity this potential is provided by a vortex via the Zeeman term. With this background, we shall now extend this model by coupling it to Ginzberg-Landau theory. The new model is expressed by the following action

$$\mathcal{L} = -\frac{1}{4}F_{ij}^2 - \frac{\rho_M}{2}\left(\nabla n^i\right)^2 - \frac{\rho_s}{2m}(\boldsymbol{\nabla}\theta - q\boldsymbol{A})^2 - \mu n^i \varepsilon^{ijk}\partial^j A^k - D\varepsilon^{ijk}n^i\partial^j n^k. \tag{27}$$

The Euler-Lagrange equation for $F$ is given by

$$\nabla^2 F - \frac{1}{2\rho^2}\sin 2F + \frac{D}{\rho_M \rho}(\cos 2F - 1) - \frac{\mu}{\rho_M}\left(\sin F B^z + \cos F \sin \phi B^x - \cos F \cos \phi B^y\right) = 0. \tag{28}$$

Even in this case we shall consider the field $F$ to be a function of $\rho$ only. This is because the Euler-Lagrange for $\Phi$ (Eq. (23)) remain unchanged in presence of a vortex along $z$ axis and thus the solution to the field equation for the field $\Phi$ is $\Phi = \phi \pm \frac{\pi}{2}$. We shall now proceed in a similar way discussed in Sec II. For a $z$ directional vortex configuration and in the first order approximation for the spin-spin interaction term we have the following decoupled equation

$$\nabla^2 F - \frac{1}{2\rho^2}\sin 2F + \frac{D}{\rho_M \rho}(\cos 2F - 1) - N\frac{2\mu}{\lambda^2 q\rho_M}\sin F K_0(\frac{\rho}{\sqrt{2\pi}\lambda}) = 0. \tag{29}$$

The last term in the above equation due to the vortex provides the potential term needed for skyrmion solution. However, as vortex is a localized object this term decays with distance very rapidly($\sim K_0\left(\frac{\rho}{\sqrt{2\pi}\lambda}\right)$) and therefore

skyrmion-like spin configuration is limited only up to a length scale which is determined by penetration depth and the ratio $\dfrac{2\mu}{q\rho_M}$. Outside this length scale the solution exhibit spiral wave like configuration which is a feature of presence of DMI term in absence of the potential terms. The numerical solution for the choice $\lambda = 4 \text{ eV}^{-1}$, $\dfrac{D}{\rho_M} = 1 \text{ eV}$, $\dfrac{2\mu}{q\rho_M} = 100$ and $N = -10$ is shown in Fig 6. Thus the solution in presence of DMI shows very similar qualitative features to previous solutions with out DMI.

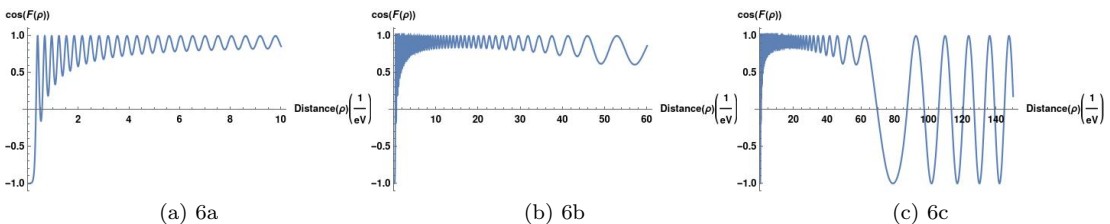

(a) 6a  (b) 6b  (c) 6c

FIG. 6. Skyrmion solution with DMI interaction

## IV. SPIN WAVES IN NLS+GL MODEL

In the previous sections we have shown that in our model a skyrmion like spin configuration arises within a length scale determined by penetration depth and the ratio of coupling constants, as well as the winding number of vortex configuration. However, this is a static configuration of the unit spin field $\hat{n}$. In general, the spin configurations in such system can be time dependent. As mentioned in the introduction, the study of spin waves in such systems provides a possible way to probe their magnetic features. In this section we shall consider the spin field to be time dependent and try to derive an exact propagating wave solution from the Euler-Lagrange equation, both in the absence and presence of vortices. We start from the following action for the NLS+GL model with time dependent spin field $\hat{n}(t)$

$$\mathcal{L} = -\frac{1}{4}F_{\mu\nu}^2 + M_0 \boldsymbol{C}(\hat{n}) \cdot \partial_t \hat{n} - \frac{\rho_M}{2}\left(\boldsymbol{\nabla} n^i\right)^2 + \frac{\rho_s}{2m}(\partial_\mu \theta - qA_\mu)^2 - \mu n^i(\varepsilon^{ijk}\partial^j A^k + \delta^{iz}B_0) + \rho_M Q\,\varepsilon^{ijk}n^i\partial^j n^k. \quad (30)$$

where we have written the DMI coefficient $D = \rho_M Q$ and $\boldsymbol{C}(\hat{n})$ is introduced as the conjugate momentum for the field $\hat{n}(t)$, and $M_0$ is the spin per unit volume as defined earlier, $M_0 = S\,a^{-3}$. Also, $\boldsymbol{B}_0 \equiv B_0\hat{z}$ is the effective background magnetic field due to spontaneous magnetization existing in ferromagnet. The equation of motion of the field $\hat{n}$ from the above action is given by

$$M_0\left(\partial_{n_j}C_i - \partial_{n_i}C_j\right)\partial_t n_i + \rho_M\nabla^2 n_j + 2\rho_M Q\,\varepsilon_{jkl}\partial_k n_l + \mu(B_j + \delta_{jz}B_{0j}) = 0. \quad (31)$$

Here we have denoted by $\partial_{n_i}$ a partial derivative with respect to $n_i$. The conjugate vector field $\boldsymbol{C}$ obeys [28, 32]

$$\left(\partial_{n_i}C_j - \partial_{n_j}C_i\right) = \varepsilon_{ijk}n_k\,, \quad (32)$$

using which we get the Landau-Lifshitz equation

$$M_0\partial_t\hat{n} = \rho_M\,\hat{n}\times\nabla^2\hat{n} + 2\rho_M Q\,(\hat{n}\cdot\boldsymbol{\nabla})\,\hat{n} + \mu\hat{n}\times(\boldsymbol{B}+\boldsymbol{B}_0)\,. \quad (33)$$

To derive the spin wave solution we shall solve this equation along with the wave equation for magnetic field

$$\left(\partial_t^2 - \nabla^2 + \frac{q^2\rho_s}{m}\right)\boldsymbol{B} = \frac{q\rho_s}{m}\boldsymbol{\nabla}\times(\boldsymbol{\nabla}\theta) + \mu\boldsymbol{\nabla}\times(\boldsymbol{\nabla}\times\hat{n}). \quad (34)$$

### A. Spin wave dispersion in absence of vortex

We shall first try to solve for the case where no vortex is present i.e. $\boldsymbol{\nabla}\times(\boldsymbol{\nabla}\theta) = 0$. For the spin waves propagating along $\hat{z}$ we take the following ansatz

$$\hat{n} = \sin\alpha\left[\cos\left((k+Q)z - \omega t\right)\hat{x} + \sin\left((k+Q)z - \omega t\right)\hat{y}\right] + \cos\alpha\hat{z}. \quad (35)$$

Here we have taken the wave to be circularly polarized and have chosen the polarization to be right handed one. We shall continue with this notation through all subsequent sections. One can find out the expression of the magnetic field for this spin wave ansatz using the method of Green functions,

$$B^i = \int d^4x' \; G(x - x') \left[ -\mu \nabla^2 n^i \right] (x').$$ (36)

Using Fourier transform, one can write the Green function as

$$G(x - x') = \int \frac{d^4p}{(2\pi)^4} \frac{1}{-p_0^2 + p^2 + \tilde{m}^2} e^{i(p_0(t-t') - \boldsymbol{p} \cdot (\boldsymbol{x} - \boldsymbol{x}'))},$$ (37)

where we have written $\tilde{m} = \dfrac{q^2 \rho_s}{m}$ as before. We get an expression for $B^x$ from this,

$$B^x = \frac{\mu \sin \alpha (k + Q)^2}{-\omega^2 + (k + Q)^2 + \tilde{m}^2} \cos \left( -\omega t + (k + Q)z \right),$$ (38)

and an expression for $B^y$ in a similar manner,

$$B^y = \frac{\mu \sin \alpha (k + Q)^2}{-\omega^2 + (k + Q)^2 + \tilde{m}^2} \sin \left( -\omega t + (k + Q)z \right).$$ (39)

However, as $\nabla^2 n^z = 0$, there is no $z$ component of $\boldsymbol{B}$. Therefore the total solution can be written as

$$\boldsymbol{B} = \frac{\mu(k + Q)^2}{-\omega^2 + (k + Q)^2 + \tilde{m}^2} \hat{n} - \frac{\mu \cos \alpha (k + Q)^2}{-\omega^2 + (k + Q)^2 + \tilde{m}^2} \hat{z}.$$ (40)

With this expression in hand, one can now put the ansatz of Eq. (35) for the spin field into Eq. (33) and find the following equation involving the frequency $\omega$ and wave vector $k$,

$$\omega - \omega_0 - a \left( k^2 - Q^2 \right) + \frac{b(k + Q)^2}{-\omega^2 + (k + Q)^2 + \tilde{m}^2} = 0,$$ (41)

where we have written $a = \dfrac{\rho_M \cos \alpha}{M_0}$, $b = \dfrac{\mu^2 \cos \alpha}{M_0}$, $\omega_0 = \dfrac{\mu B_0}{M_0}$. Clearly, the last term of Eq. (41) is the contribution from the dynamically generated magnetic field and therefore is in general dependent on $\omega, k$. Thus in absence of this contribution we recover the usual dispersion relation for spin waves in a ferromagnet: $\omega = \omega_0 + a \left( k^2 - Q^2 \right)$.

To understand the physical meaning of the above cubic equation we shall try to derive approximate solutions corresponding to different regions of the frequency of spin waves. For that we shall compare the frequency of the spin waves with the characteristic scale of the superconductor i.e. the photon mass $\tilde{m}$. First we consider the low frequency limit in which frequency is much lower than photon mass $\tilde{m}$, i.e. $\omega^2 \ll \tilde{m}^2$. In this approximation we get

$$\begin{aligned} \omega - \omega_0 &= a \left( k^2 - Q^2 \right) - \frac{b(k + Q)^2}{-\omega^2 + (k + Q)^2 + \tilde{m}^2} \\ &= a \left( k^2 - Q^2 \right) - \frac{b(k + Q)^2}{(k + Q)^2 + \tilde{m}^2} \left[ 1 - \frac{\omega^2}{(k + Q)^2 + \tilde{m}^2} \right]^{-1} \\ &\simeq a \left( k^2 - Q^2 \right) - \frac{b(k + Q)^2}{(k + Q)^2 + \tilde{m}^2} + \cdots \end{aligned}$$ (42)

If we ignore other terms in the expansion and take the case $Q = 0$, $\cos \alpha > 0$, we get the result obtained by Braude and Sonin [26]. This dispersion relation has a particular feature that the minimum in the spin wave dispersion curve appear at some finite value of momentum $k = k_{Q=0} = k_0$. Therefore around that point two different spin wave mode exist: one with positive group velocity and another with negative group velocity. This is a distinctive feature for the ferromagnetic superconductors. In presence of a DMI term with small value of $Q$ this minimum changes. If we denote the $k$ value at which the minimum occurs for a DMI coefficient $Q$ as $k_Q$, it is easy to see that $k_Q$ satisfies the equation

$$k_Q \left[ (k_Q + Q)^2 + \tilde{m}^2 \right]^2 = \frac{\mu^2}{\rho_M} (k_Q + Q) \tilde{m}^2.$$ (43)

From the above equation we can see that at $Q = 0$ we have three real solutions : $k_0 = 0$ and $k_0 = \pm\tilde{m}\sqrt{\frac{1}{\tilde{m}}\sqrt{\frac{\mu^2}{\rho_M} - 1}}$.

It was shown in [26] that for $\sqrt{\frac{\mu^2}{\rho_M}} < \tilde{m}$ there is a minimum at $k = 0$, while for $\sqrt{\frac{\mu^2}{\rho_M}} > \tilde{m}$ the minimum shifts to $k = k_0$. This is depicted in Fig 7a and 7b for two different values of $\omega_0$ and $\tilde{m} = 10$, $Q = 0$. The wave stops propagating when the frequency goes to zero, so that for a sufficiently small $\omega_0$, the spin wave disappears for some values of $k$, leading to a break in the spectrum. The spin waves with $k < k_0$ will propagate with a negative group velocity while for the spin waves with $k > k_0$ the group velocity is positive. In presence of a DMI term one of the two minima has lower energy, depending on the sign of $Q$, as shown in Fig 7c and 7d where we have taken $\tilde{m} = 10$ eV, $a = 0.06$ eV$^{-1}$, $b = 10$ eV, $|Q| = 0.4$ eV. In the small $Q$ approximation the location of two minimum points of dispersion curve can be written as

$$k_Q \approx \pm(k_0 - Q) + \cdots \tag{44}$$

Thus we can see that in presence of DMI term the location of $k_0$ gets shifted by $Q$. A similar analysis can be done for $\cos\alpha < 0$. In this case, depending on whether $\frac{b}{a}$ is smaller or larger than $\tilde{m}^2$ there can be one or two maxima.

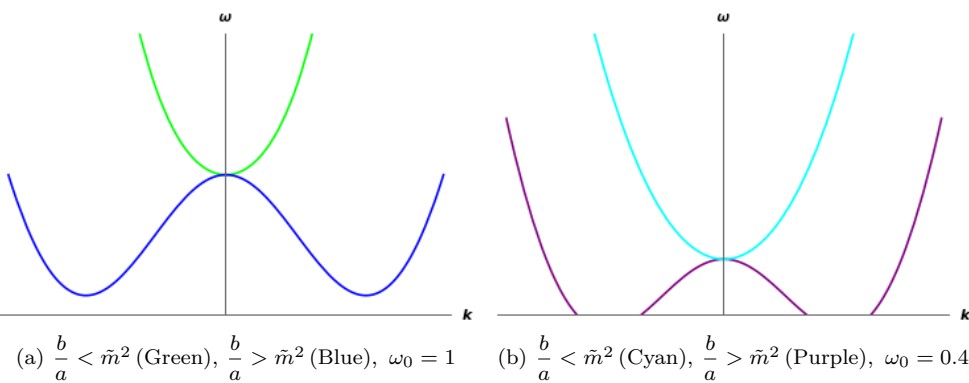

(a) $\frac{b}{a} < \tilde{m}^2$ (Green), $\frac{b}{a} > \tilde{m}^2$ (Blue), $\omega_0 = 1$     (b) $\frac{b}{a} < \tilde{m}^2$ (Cyan), $\frac{b}{a} > \tilde{m}^2$ (Purple), $\omega_0 = 0.4$

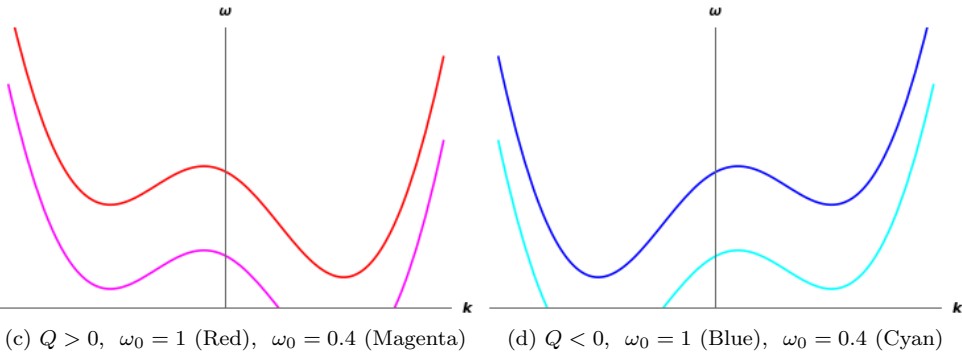

(c) $Q > 0$, $\omega_0 = 1$ (Red), $\omega_0 = 0.4$ (Magenta)     (d) $Q < 0$, $\omega_0 = 1$ (Blue), $\omega_0 = 0.4$ (Cyan)

FIG. 7. Dispersion of spin waves at different values of $a$, $b$, $Q$, and $\tilde{m}$, for $\omega \ll \tilde{m}$

Next let us consider the region where the frequency is close to the photon mass, i.e. $|\omega - \tilde{m}|$ is small compared to

$\omega$ or $\tilde{m}$. In this case one can expand the cubic equation as

$$\begin{aligned}
\omega - \omega_0 &= a\left(k^2 - Q^2\right) - \frac{b(k+Q)^2}{-\omega^2 + (k+Q)^2 + \tilde{m}^2} \\
&= a\left(k^2 - Q^2\right) - b\left[1 - \frac{\omega^2 - \tilde{m}^2}{(k+Q)^2}\right]^{-1} \\
&\simeq a\left(k^2 - Q^2\right) - b\left[1 + \frac{\omega^2 - \tilde{m}^2}{(k+Q)^2}\right] + \cdots.
\end{aligned} \tag{45}$$

The solution of this quadratic equation is

$$\omega = -\frac{(k+Q)^2}{2b} \pm \sqrt{\tilde{m}^2 + \frac{(k+Q)^4}{b^2} - (k+Q)^2\left[1 - \frac{\omega_0}{b} - \frac{a}{b}(k^2 - Q^2)\right]}. \tag{46}$$

In Fig. 8 we have displayed a particularly interesting case of this solution, which describes the dispersion in presence of superconductivity, with $\tilde{m} = 4$, $\dfrac{b}{a} \sim 1$ and with the DMI coefficient $Q$ taken to be $Q = 0, 0.1, -0.1$. At $Q = 0$ the dispersion curve describes two degenerate minima at some non-zero value of wave vector $k$. As we start to increase $Q$ from 0 the degeneracy breaks and two non-degenerate minima develop on the dispersion curve.

The excitation around the local minimum with a finite gap describes a roton like mode. Such roton like character in spin waves was proposed theoretically for different magnetic systems and termed magnetic rotons [33, 34]. Thus we can say that the spin wave excitations near to these minimum shown in Fig. 7 and Fig. 8 describes magneto roton excitations which is a very interesting feature.

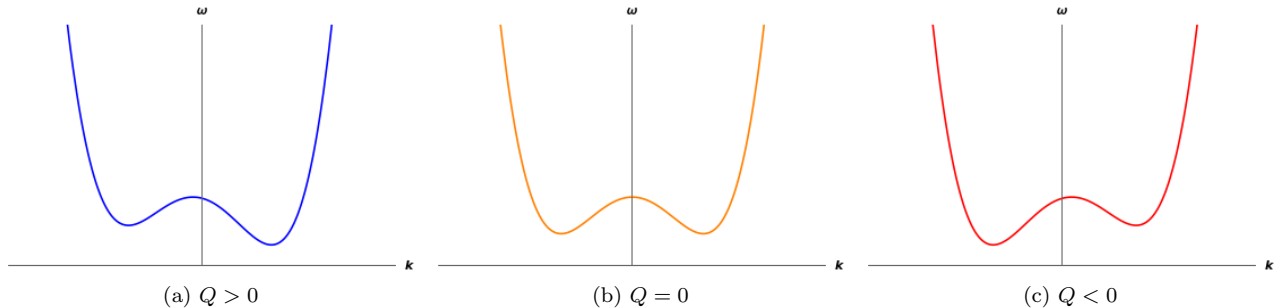

| (a) $Q > 0$ | (b) $Q = 0$ | (c) $Q < 0$ |

FIG. 8. Dispersion of spin waves at different values of $a$, $b$, $Q$, $\tilde{m}$ for $\omega \sim \tilde{m}$.

Lastly, one can take the frequency of the spin waves to be higher than the photon mass i.e. $\omega^2 \gg \tilde{m}^2$. In this limit effect of superconductivity on the the spin wave spectrum will be suppressed and nature of spin waves would be similar to that in a ferromagnet coupled to electromagnetic fields. The equation governing the behaviour of spin waves in this regime is given below

$$\omega - \omega_0 - a\left(k^2 - Q^2\right) + \frac{b(k+Q)^2}{-\omega^2 + (k+Q)^2} = 0. \tag{47}$$

The nature of the spin waves in a ferromagnet with Zeeman coupling is a well studied subject and therefore we do not discuss that here.

### B.   Effect of a static vortex on spin wave solution

Let us now introduce vortices in the theory. These act as a source of an inhomogeneous magnetic field. Due to this inhomogeneous field the amplitude of the spin field cannot remain constant through out the system. In particular, in the presence of a cylindrically symmetric Abrikosov vortex along the $Z$ axis, a radially varying magnetic field profile is created, decaying sharply with distance. In such a case, let us take the following ansatz for the spin wave

$$\hat{n} = \sin \alpha(\rho)\left[\cos\left((k+Q)z - \omega t\right)\hat{x} + \sin\left((k+Q)z - \omega t\right)\hat{y}\right] + \cos \alpha(\rho)\hat{z}, \tag{48}$$

where $\rho$ is the radial distance from the axis, while the frequency and wave vector are assumed to be constant. The profile of the function $\alpha(\rho)$ has to be determined from the Landau-Lifshitz equation of motion for $\hat{n}$. For this let us again write down again the Landau-Lifshitz equation (49)

$$\partial_t \hat{n} = \rho_M \, \hat{n} \times \nabla^2 \hat{n} + 2\rho_M \, Q \, (\hat{n} \cdot \boldsymbol{\nabla}) \, \hat{n} + \mu \hat{n} \times \boldsymbol{B} \,. \tag{49}$$

As the DMI term will bring a dependence on the angular coordinate $\phi$, in presence of this term solution will be much harder to derive. For simplicity, we set $Q = 0$ and proceed to solve the equation. We showed earlier that the magnetic field due to a cylindrically symmetric Abrikosov vortex has the profile

$$B^z = \frac{1}{2\pi\lambda^2 q} \int d^3x' \, G(x - x')\Sigma^z(x') = \frac{2N}{\lambda^2 q}K_0\left(\frac{\rho}{\sqrt{2\pi}\lambda}\right). \tag{50}$$

With this magnetic field profile for a vortex, Eq. (49) can be rewritten as

$$\partial_t \hat{n} = \rho_M \hat{n} \times \, \nabla^2 \hat{n} + \frac{2\mu N}{\lambda^2 q}K_0\left(\frac{\rho}{\sqrt{2\pi}\lambda}\right) \, \hat{n} \times \hat{z} - \mu^2 \hat{n} \times \int d^4x' G(x - x') \, \nabla^2 \hat{n}(x') \,. \tag{51}$$

Putting the ansatz of Eq. (48) into Eq. (51) we obtain for the field $\alpha(\rho)$

$$\partial_\rho^2 \alpha + \frac{1}{\rho}\partial_\rho \alpha - \frac{k^2}{2}\sin(2\alpha(\rho)) + \omega \sin\alpha(\rho) - \frac{2\mu N}{\lambda^2 q}K_0\left(\frac{\rho}{\sqrt{2\pi}\lambda}\right)\sin\alpha + \mu^2 \zeta\,(\omega, k, \rho) = 0 \,, \tag{52}$$

where we have written

$$\zeta\,(\omega, k, \rho) = \int d^2x'_\parallel d^2p_\parallel \, e^{i\mathbf{p}_\parallel \cdot (\mathbf{x}' - \mathbf{x})_\parallel} \left[\cos\alpha(\rho)f_1(\rho', \omega, k, \tilde{m}) + \sin\alpha(\rho)f_2(\rho, \tilde{m})\right],$$

$$f_1(\rho', \omega, k, \tilde{m}) = \frac{\left[\cos\alpha(\rho')\left(\partial_{\rho'}^2 \alpha(\rho') + \frac{1}{\rho'}\partial_{\rho'}\alpha(\rho')\right) - \sin\alpha(\rho')(\partial_{\rho'}\alpha(\rho'))^2 - \frac{k^2}{2}\sin(\alpha(\rho'))\right]}{\mathbf{p}_\parallel^2 + k^2 + \tilde{m}^2 - \omega^2}, \tag{53}$$

$$f_2(\rho', \tilde{m}) = \frac{\left[\sin\alpha(\rho')\left(\partial_{\rho'}^2 \alpha(\rho') + \frac{1}{\rho'}\partial_{\rho'}\alpha(\rho')\right) + \cos\alpha(\rho')(\partial_{\rho'}\alpha(\rho'))^2\right]}{\mathbf{p}_\parallel^2 + \tilde{m}^2}.$$

Also, the symbols $x_\parallel$, $p_\parallel$ are the $x$, $y$ components of the position and momentum vectors. The field $\zeta(\omega, k, \rho)$ is the magnetic field generated by the spin field $\hat{n}$ and has to be determined in a self-consistent way. Eq. (52) governs the behaviour of the field $\alpha(\rho)$ for a given constant frequency $\omega$ and wave number $k$. As we can see, this equation resembles Eq. (17) where we had not taken $\mu^2$ term for simplicity. Thus the amplitude of the spin wave give rise to a spatial profile similar to the skyrmion solution shown previously. In particular, the variation of the amplitude of spin wave near the vortex and far from it would be different which would be an interesting observable feature.

## V.  DISCUSSION AND OUTLOOK

In this work we have considered a nonlinear sigma model(NLSM) coupled to a Ginzburg Landau (GL) model of superconductivity in the Landau limit via a Zeeman coupling term. Such a model would describe a ferromagnetic superconductor and topological excitation that may appear in such systems. Here we have analysed how topological structures like vortices in superconducting order and skyrmion in the ferromagnetic order are intertwined. We have shown that for a straight vortex along the $z$ axis of the coordinate system a skyrmion like configuration arise with in a finite region whose size depends on the London penetration depth of the superconductor and also on other parameters of the model.

This suggest that in the model a coupled structure of skyrmion and vortex indeed appears. In addition, we have shown that this particular solution arises only when the spin at the origin aligns along the magnetic field of the vortex. This suggest that such system, if stabilized, can manifest a coupled motion of the skyrmion and vortices. It is known for a long time that in presence of an external current $\mathbf{J}_T$ a moving vortex with velocity $\mathbf{V}_L$ experience a force transverse to the direction of it's velocity which is given by[35]

$$F_L = \pi\hbar(n_\infty - n_0)\hat{\mathbf{z}} \times \mathbf{V}_L, \tag{54}$$

where $n_\infty$ and $n_0$ are the particle densities far outside the vortex and at the core respectively. This force is independent of charge and not of electromagnetic origin. For a vortex coupled to a skyrmion in the manner described in this paper

this transverse force will produce a motion of the vortex which in turn will produce a motion of the skyrmion perpendicular to the applied current. The motion of a skyrmion perpendicular to the applied current is famously known as skyrmion Hall effect[25, 36] which is recently discovered[37]. In that context skyrmion Hall effect appears due to the torque applied by the conduction electron spin via a s-d type exchange interaction of the local spins and conduction electron spin[36]. However, in the present case the skyrmion Hall effect may be produced by the coupling between vortex and skyrmion structures. This new type of skyrmion Hall effect, if found, would be a manifestation of the coupling between vortex and skyrmion like structures.

At this point we remember that while solving the equations for the function $F(\rho)$ we have always considered cylindrical symmetry (the vortex along $z$ also bears cylindrical symmetry). For this, the solutions will be valid for any plane parallel to $x - y$ plane. Thus our solution describes a cylindrical symmetric 3D structure. In practice, such a 3D system where local spin field interacts with the superconductor via a Zeeman term is difficult to stabilize. For this in most of the related works authors have considered a hybrid system where superconductor and ferromagnet exist in different bulk and interacts only via the interface of the hybrid via the Zeeman or proximity coupling. However, the field theoretic action that we have considered can also be applicable to such a situation with suitable modification (considering boundary effect etc.). Based on the analysis done in this work it is expected that a vortex induced in the superconductor in such hybrid system will couple to skyrmion structure in the ferromagnet by the Zeeman interaction. We shall try to extend our model for this hybrid structure in a later work.

Spin wave solution in such model also show some very interesting features as we had discussed. In particular roton like gapped excitation appear around the local minimum in dispersion curve. This may resemble mass generation for spin waves. Anderson-Higgs(AH) mass generation of magnons is very interesting and was recently discussed in a superconductor-ferromagnet-superconductor hetero structure [38]. Thus, as a possible future development, one can construct a field theoretic description of AH mass generation for magnons starting from a microscopic theory of our system and using effective bosonic representations.

We had mentioned that as soon as frequency of the spin waves approaches zero it becomes non-propagating, leading to a discontinuity in the dispersion curve (Fig. 7). With the approximations we had taken in Eq. (42) the $\omega - k$ curve crosses zero of $\omega$ maximum two times as can be seen. However, one can consider other higher order terms which would lead to multiple such crossings. This would create a band like formation in $\omega - k$ diagram.

The spin wave dispersion i.e. the $\omega - k$ relation, comes by solving a cubic equation Eq. (41) in the absence of a vortex — a more general equation needs to be solved in the presence of a vortex. In general, this differs from the quadratic dispersion which is the usual case. A change in the $\omega - k$ relation can lead to a modification of the thermodynamic property of a collection of magnonic excitations. We know that magnons influence the electronic specific heat as well as thermal and electrical conductivity of magnetic systems [39]. Thus a change in nature of spin waves may affect these properties which would be experimentally observable. This also opens a possible direction for future investigation.

## ACKNOWLEDGMENT

This work was partially carried out during a research visit by SM to the Leibniz Institute supported by an MoU between the S. N. Bose Centre and the Leibniz Institute. The authors gratefully acknowledge many discussions with Flavio Nogueira which provided many insightful ideas which proved to be essential for the progress and completion of this project. SM also acknowledges discussions with Arijit Haldar, Shibendu G. Choudhury, and Ramesh Pramanik.

## APPENDIX: ALTERNATIVE DERIVATION OF EQ 17 VIA DUAL FORMULATION

To proceed towards deriving the dual description let us write down the generating functional $\mathcal{Z}$ of our system

$$
\mathcal{Z} = \int \mathcal{D}n^i \mathcal{D}A^i \mathcal{D}\theta \exp\left[i \int d^4x \mathcal{L}\right],
$$
$$
\mathcal{L} = -\frac{1}{4}F_{ij}^2 - \frac{\rho_M}{2}\left(\partial_\mu n^i\right)^2 - \frac{\rho_s}{2m}(\boldsymbol{\nabla}\theta - q\boldsymbol{A})^2 - \mu n^i \varepsilon^{ijk}\partial^j A^k
$$

(55)

To proceed we shall linearize the third term in the Lagrangian by Hubbard-Stratonovich transformation as follows.

$$
\mathcal{Z} = \int \mathcal{D}n^i \mathcal{D}A^i \mathcal{D}\theta \mathcal{D}f^i \exp\left[i \int d^4x \mathcal{L}\right],
$$

$$
\mathcal{L} = -\frac{1}{4}F_{ij}^2 - \frac{\rho_M}{2}\left(\partial_\mu n^i\right)^2 - 2\sqrt{\frac{\rho_s}{2m}}f^i(\boldsymbol{\nabla}\theta - q\boldsymbol{A})^i - f^i f^i - \mu n^i \varepsilon^{ijk}\partial^j A^k,
$$

(56)

where $f^i$ is a new auxiliary vector field. The phase field $\theta$ is multivalued in presence of a vortex. So one can breakdown $\boldsymbol{\nabla}\theta$ into two different parts: $\boldsymbol{\nabla}\theta = \boldsymbol{\nabla}\theta^r + \boldsymbol{\nabla}\theta^s$, where the first part is irrotational while the second part is constituted of multivalued field $\theta^s$ and has non-zero curl. Also, the fields $\theta^r$ and $\theta^s$ behaves as independent variables. With this decomposition one can write the above partition function as

$$
\mathcal{Z} = \int \mathcal{D}n^i \mathcal{D}A^i \mathcal{D}\theta^r \mathcal{D}\theta^s \mathcal{D}f^i \exp\left[i \int d^4x \mathcal{L}\right],
$$

$$
\mathcal{L} = -\frac{1}{4}F_{ij}^2 - \frac{\rho_M}{2}\left(\partial_\mu n^i\right)^2 - 2\sqrt{\frac{\rho_s}{2m}}f^i \partial^i \theta^r - 2\sqrt{\frac{\rho_s}{2m}}f^i(\boldsymbol{\nabla}\theta^s - q\boldsymbol{A})^i - f^i f^i - \mu n^i \varepsilon^{ijk}\partial^j A^k,
$$

(57)

One can now integrate out $\theta^r$ functionally to obtain the functional delta function $\delta(\boldsymbol{\nabla}\cdot\boldsymbol{f})$. This imposes the constraint that the vector field $\boldsymbol{f}$ is conserved. This can be satisfied by the following ansatz

$$
\boldsymbol{f} = \frac{1}{2}\boldsymbol{\nabla}\times\boldsymbol{C},
$$

(58)

where $\boldsymbol{C}$ is a new vector field. This ansatz will help us rewriting the delta function as $\delta(\boldsymbol{f} - \frac{1}{2}\boldsymbol{\nabla}\times\boldsymbol{C})$. One can now integrate over $f^i$ functionally to get the following action

$$
\mathcal{Z} = \int \mathcal{D}n^i \mathcal{D}A^i \mathcal{D}\theta^s \mathcal{D}C^i \exp\left[i \int d^4x \mathcal{L}\right],
$$

$$
\mathcal{L} = -\frac{1}{4}F_{ij}^2 - \frac{\rho_M}{2}\left(\partial_\mu n^i\right)^2 - \sqrt{\frac{\rho_s}{2m}}(\boldsymbol{\nabla}\times\boldsymbol{C})\cdot\boldsymbol{\nabla}\theta^s + q\sqrt{\frac{\rho_s}{2m}}(\boldsymbol{\nabla}\times\boldsymbol{C})\cdot\boldsymbol{A} - \frac{1}{4}(\boldsymbol{\nabla}\times\boldsymbol{C})^2 - \mu n^i \varepsilon^{ijk}\partial^j A^k,
$$

(59)

Let us now define vorticity in the following manner

$$
\boldsymbol{\Sigma} = \frac{\Phi_0}{2\pi}\boldsymbol{\nabla}\times(\boldsymbol{\nabla}\theta^s).
$$

(60)

With this definition and recalling that $\sqrt{\dfrac{\rho_s}{2m}} = \dfrac{1}{\sqrt{4\pi}\lambda}\dfrac{\Phi_0}{2\pi}$, $\lambda$ being the penetration depth, we can rewrite the generating functional

$$
\mathcal{Z} = \int \mathcal{D}n^i \mathcal{D}A^i \mathcal{D}\theta^s \mathcal{D}C^i \exp\left[i \int d^4x \mathcal{L}\right],
$$

$$
\mathcal{L} = -\frac{1}{4}F_{ij}^2 - \frac{\rho_M}{2}\left(\partial_\mu n^i\right)^2 - \frac{1}{\sqrt{4\pi}\lambda}\boldsymbol{C}\cdot\boldsymbol{\Sigma} + \frac{1}{\sqrt{4\pi}\lambda}(\boldsymbol{\nabla}\times\boldsymbol{C})\cdot\boldsymbol{A} - \frac{1}{4}(\boldsymbol{\nabla}\times\boldsymbol{C})^2 - \mu n^i \varepsilon^{ijk}\partial^j A^k.
$$

(61)

To obtain a theory written in terms of vorticity ($\boldsymbol{\Sigma}$) and unit vector field $\hat{n}$ we integrate out both the vector fields $A^i$ and $C^i$ and obtain the following partition function

$$
\mathcal{Z} = \int \mathcal{D}n^i \mathcal{D}\theta^s \exp\left[i \int d^4x \mathcal{L}\right],
$$

$$
\mathcal{L} = -\frac{\rho_M}{2}\left(\partial_\mu n^i\right)^2 - \frac{\mu}{2\pi\lambda^2}\hat{n}\cdot\left(\frac{1}{\nabla^2 - \tilde{m}^2}\boldsymbol{\Sigma}\right) + \frac{1}{2\pi\lambda^2}\boldsymbol{\Sigma}\cdot\left(\frac{1}{\nabla^2 - \tilde{m}^2}\boldsymbol{\Sigma}\right) + \mu^2\hat{n}\cdot\left(\frac{\nabla^2}{\nabla^2 - \tilde{m}^2}\hat{n}\right),
$$

(62)

where we have written $\tilde{m}^2 = \frac{1}{2\pi\lambda^2}$. This is the dual theory we wanted to achieve. We shall now show that the Eq (17) can be easily obtained from the above dual theory. For that we first put the ansatz of Eq (3) and get the following Lagrangian

$$
\mathcal{L} = \frac{\rho_M}{2}\left(\partial_i F \partial_i F + \frac{1}{\rho^2}\sin^2 F\right) - \frac{\mu}{2\pi\lambda^2}\hat{n}\cdot\left(\frac{1}{\nabla^2 - \tilde{m}^2}\boldsymbol{\Sigma}\right),
$$

(63)

where we have ignored the vortex vortex interaction term and spin-spin interaction term which is a higher derivative term. The Euler-Lagrange equation for the field $F$ is given by

$$\nabla^2 F - \frac{\sin 2F}{2\rho^2} + \frac{\mu}{2\pi\rho_M\lambda^2}\sin F\left(\frac{1}{\nabla^2 - \tilde{m}^2}\Sigma^z\right)$$
$$- \frac{\mu}{2\pi\rho_M\lambda^2}\cos F\cos\phi\left(\frac{1}{\nabla^2 - \tilde{m}^2}\Sigma^x\right) - \frac{\mu}{2\pi\rho_M\lambda^2}\cos F\sin\phi\left(\frac{1}{\nabla^2 - \tilde{m}^2}\Sigma^y\right) = 0. \tag{64}$$

For a vortex along $z$ axis we can write from the above equation

$$\nabla^2 F - \frac{\sin 2F}{2\rho^2} - \frac{\mu}{2\pi\rho_M\lambda^2}\sin F\left(\frac{1}{-\nabla^2 + \tilde{m}^2}\Sigma^z\right) = 0. \tag{65}$$

One can easily understand that

$$\left(\frac{1}{-\nabla^2 + \tilde{m}^2}\Sigma^z\right) = N\,\Phi_0\int d^3y\,dz'\,G(\boldsymbol{x} - \boldsymbol{y})\int dz'\delta^3(\boldsymbol{y} - \hat{z}\,z') = 2N\,\Phi_0\,K_0(\tilde{m}\rho) \tag{66}$$

Thus finally we obtain

$$\nabla^2 F - \frac{\sin 2F}{2\rho^2} - N\frac{2\mu\Phi_0}{2\pi\rho_M\lambda_0^2}\sin F\,K_0(\tilde{m}\rho) = 0. \tag{67}$$

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
