# Peer review of "Skyrmion-vortex hybrid and spin wave solutions in ferromagnetic superconductors"

_SciPost Physics, doi:SciPost Phys. 19, 022 (2025)_

## Round 1 · Referee Report · Anonymous (Referee 1) · 2024-11-16

Strengths

In this paper, the authors Mukherjee and Lahiri discuss aspects of composite magnetic skyrmion and superconducting vortex excitations in ferromagnetic superconductors. The authors employ an effective Ginzburg-Landau type of approach to study such coupled excitations both in the presence and absence of Dzyaloshinskii-Moriya interaction. The first part of this study focuses on the properties of an individual magnetic skyrmion, such as, the size of its radius as a function of parameters characterizing the superconducting condensate, e.g., the penetration depth. In the second part of this work, the authors study the magnon dispersion through such a ferromagnetic superconductor. A brief discussion of the effects of a superconducting vortex on the magnon propagation is also provided.

  1. The authors discuss a timely topic, since there has been interest recently on such composite excitations due to their potential role in creating Majorana excitations and implementing quantum computing.

  2. One novelty of the authors’ approach is that they explored the magnon dispersion in ferromagnetic superconductors with Dzyaloshinskii-Moriya interaction.

  3. The technical approach appears correct.

  4. The results appear to be reasonable.

  5. Overall this manuscript reads well.

Weaknesses

  1. The authors claim that they discuss novel skyrmion and magnon solutions in ferromagnetic superconductors, but I have reservations when it comes to the degree of novelty of this work.

  2. As the authors discuss in their introduction, there exist a number of works which have recently studied various properties of such excitations, especially when it comes to interfaces and heterostructures of ferromagnets and superconductors. The type of analysis followed by the authors and the questions that they try to answer do not focus on the nature of systems that lead to these composite pairs of excitations. For instance, in ferromagnetic superconductors, one expects that ferromagnetism and superconductivity are competing if they come from the same electrons. As the authors discuss, in many rare earth superconductors these two phases of matter stem from different degrees of freedom which could make their coexistence more feasible.

Here the authors simply assume the presence of ferromagnetism/skyrmions and spin-singlet superconductivity, without trying to describe the regime in which such a coexistence is viable. This would be an interesting question to answer and would allow this work to stand out compared to previous studies.

  1. Along the same lines, the authors use some “typical” numbers for the penetration depth, i.e., λ~1/eV , or a few microns as they say, without providing any explanation regarding how they obtain this number. What are the systems that the authors have in mind? While I understand that a Ginzburg-Landau analysis relies on symmetry and is useful precisely for its generality, it is important for the reader to know what comes out of this analysis from the representative systems. Therefore, what is missing here, is more of a clearer target of systems that this work plans to address.

  2. Aside from the above, the actual analysis of the properties of a magnetic skyrmion in a vortex of a ferromagnetic superconductor yields two main, but very obvious, results. First that the radius of the magnetic skyrmion increases with the increase of the penetration depth. The second is that for a fixed magnetic skyrmion profile, the sign of the charge N of the superconducting vortex is also fixed. This is, however, an expected result. Instead of this anticipated property, the authors do not answer what is the value for the vorticity N that is stabilized for the superconducting vortex. Does one always have |N|=1, or other values are also possible? In addition to that, by minimizing the energy with respect to N, the authors should be in a position to answer for what range of parameter values is such a composite excitation possible. This is one more important question that the authors did not answer in this work.

  3. The last part of the authors’ work addresses the dispersion of magnons with and without Dzyaloshinskii-Moriya interaction (DMI) in ferromagnetic superconductors. As the authors mention, the magnon dispersion has been previously studied in ferromagnets with and without DMI, and in ferromagnetic superconductors without DMI. The novel aspect of the authors’ approach is that they considered ferromagnetic superconductors with DMI. In the absence of a composite excitation, the results are straightforward and this is an interesting addition to the existing literature. However, when the authors move on to discuss the effects of a vortex, their discussion is very brief.

  4. The presentation could have been much smoother. The figures also need improvements in resolution and style.

Report

Due to all the above weaknesses that I mentioned, I unfortunately cannot recommend this work for publication in this journal. At least not in its present form. The reason is that the authors did not address fundamental aspects of the physics of such composite excitations, which would be necessary in order to claim that this work provides substantially new or important insights regarding these systems. Nevertheless, I think that this is an interesting topic and I encourage the authors to further focus on the above aspects in order to strengthen the value and impact of their work.

Requested changes

Aside from the open questions that I mentioned earlier, I append below a number of more specific comments that I believe that need to be addressed.

  1. I think that the word “Novel” in the title needs to be re-examined. In the introduction the authors write:

“the radius of the the skyrmion-like solution depends on the penetration depth and also the polarity of the skyrmion depends on the sign of the winding number. Thus our solution describes a novel topological structure- namely a skyrmion-vortex composite.”

I believe that the two features mentioned by the authors are quite general for such excitations even in proximity systems and are not sufficient to claim novelty.

  1. Throughout this work, the authors refer to the B.M coupling as the Zeeman term. Personally, I think that the Zeeman coupling refers to the coupling between an external magnetic field to the magnetic moment of a particle, but at the Hamiltonian level. The B.M coupling arises in electrodynamics in matter and does only appear due to the Zeeman mechanism. Other types of magnetoelectric couplings etc. can contribute to the coupling between B and M. Therefore, in spite of the superficial analogy, I believe that the authors should avoid any association of the term B.M to the Zeeman effect.

  2. In the Ginzburg-Landau formulation of equation 1, there are no direct couplings between magnetism and the superconducting order parameter. In earlier work by Varma et al. that was justified. What are the systems that the authors have in mind here, and do they satisfy the same conditions?

  3. The authors mention:

“However, the nature of the composite topological configuration has not been studied from a continuum field theoretic perspective as far as we know”

It is worthy to discuss more clearly the benefits of the continuum field theoretic approach compared to others. Otherwise it is not obvious why by employing this method alone would be preferred or interesting on its own.

  1. “As we shall see from the Euler-Lagrange equation, this will lead us to a new coupled structure in which the vortex is coupled to a skyrmion-like configuration.”

I think that the stabilization of a composite skyrmion-vortex excitation due to the stray fields of the magnetic component and, in turn, due to a B.M term has been previously discussed. See for instance Ref. [1] and the experiment in Ref. [15]. Therefore, I do not think that this is a new result.

  1. In contrast to proximity systems which are quasi-two-dimensional, here the authors discuss bulk three dimensional systems. In three dimensions the magnetic skyrmion solution that the authors obtain extends uniformly along one axis. Topology typically protects magnetic skyrmion in two dimensions. What renders the skyrmion tubes discussed here robust against disorder? What would happen if disorder would be also considered present?

  2. The authors mention:

“Outside that region the spin field oscillates with radial distance, which can be compared to the spiral magnetic phase described by Varma et al [23].”

From what I understand the authors refer to the oscillations in F(ρ). Due to this, the spin configuration winds in space. However, it is not obvious why the authors attribute this to the spiral phase that Varma and colleagues discussed. That phase emerges on top of a uniform superconducting density. Couldn’t these oscillation simply be a result of the defect? Do the authors imply that defects could possibly stabilize the spiral phase discussed by Varma? If the authors cannot provide a more transparent connection betweent the two phases, maybe it is better to remove such unsupported comments.

  1. “have taken the limit in which the amplitude of the superconducting order parameter is kept fixed”

I think it is better for the authors to clearly state that they essentially took the coherence length of the superconductor to zero, thus, allowing for the pairing gap to be zero only at a point, that is, the core of the vortex.

  1. “Vortex nucleation by spin textures such as Skyrmions was studied previously by Baumard et al [2] and it was also argued by Greenside et al [23] and by Ng and Varma [24] that in the SVP phase, vortices are spontaneously produced by magnetization.”

I am not convinced that these prior results can be directly applied to the case that the present authors wish to discuss. The work by Baumard referred to a heterostructure, while the remaining two works did not discuss a magnetic skyrmion. I urge the authors to look into the important issue of the feasibility of this composite excitation in the systems of their interest, after they also have more clearly defined the latter.

  1. At the end of page 5 one finds some estimates for the penetration depth. How do the author get these numbers? What superconductors do they have in mind? Can they provide a concrete example of a material?

  2. The presentation of the figures needs improvements, see for instance, figure 6.

  3. Towards the end, the authors write:

“In practice, such a 3D system where local spin field interacts with the superconductor via a Zeeman term is difficult to stabilize. For this in most of the related works authors have considered a hybrid system where superconductor and ferromagnet exist in different bulk and interacts only via the interface of the hybrid via the Zeeman or proximity coupling. However, the field theoretic action that we have considered can also be applicable to such a situation with suitable modification (considering boundary effect etc.). Based on the analysis done in this work it is expected that a vortex induced in the superconductor in such hybrid system will couple to skyrmion structure in the ferromagnet by the Zeeman interaction. We shall try to extend our model for this hybrid structure in a later work.”

I really appreciate this “self-criticism” provided by the authors. At the same time, however, this difference in approach implies that a number of aspects have remained unaddressed in this analysis, that I believe should have already been present. The authors discuss three dimensional system, but at the same time do not appear optimistic regarding its realization. They somehow try to establish a connection with heterostructures. However, this emphasizes the need for evaluating which results of this work are realistic, and which ones have more of an explorative nature which may be difficult to realize in a lab. The theory by the authors can be “extended” to quasi-two-dimensional materials, but it is important to clarify whether such a composite excitation can exist in bulk ferromagnetic superconductors, or, whether a heterostructure appears to be the best experimental route to realize these excitations in the lab.

Recommendation

Ask for major revision

---

## Round 3 · Referee Report · Anonymous (Referee 1) · 2025-5-31

Strengths

  1. The authors discuss a timely topic, since there has been interest recently on such composite excitations due to their potential role in creating Majorana excitations and implementing quantum computing.

  2. The authors investigate an interesting problem concerning superconducting electrons which are under the influence of the exchange field generated by localized moments. Such a study may find experimental relevance for rare-earth superconductors and reveals how magnetic skyrmions can be engineered in superconducting vortices.

  3. One other novelty of the authors’ approach is that they explored the magnon dispersion in ferromagnetic superconductors with Dzyaloshinskii-Moriya interaction.

  4. The technical approach appears correct.

  5. The results are reasonable and interesting.

  6. Overall this manuscript reads well.

  7. This work can inspire further studies in this field and motivate revisiting the possibility of Majorana zero modes in rare-earth superconductors. Some early ideas when it comes to Majorana modes in such systems were first discussed by Martin and Morpurgo in a PRB of 2012. There, the Majorana zero modes formed flat bands. Here, instead, the authors propose a route to realize localized Majorana modes, which can be advantageous for quantum computing applications.

  8. This version is substantially improved compared to the previous one.

Weaknesses

  1. There are still a few spots where the presentation needs to be improved. I raise these issues later on.

  2. The authors have added the minimization of the energy in order to find the vorticity value which becomes stabilized for the superconducting vortex of the composite skyrmion-vortex excitation. However, the authors minimize the energy by not properly accounting for the integer character of the vorticity. A suitable amendment is required here.

Report

I commend the authors on their efforts to rectify the first submitted version of this work. Overall, I believe that they did a good job. Unfortunately, I still find that there exist a few outstanding issues that need to be addressed before I can recommend this work for publication in this journal.

Requested changes

  1. Throughout the text, the authors use the term ferromagnetic superconductors. In contrast, in the title, they use the term superconducting ferromagnets. I suggest that the authors use "ferromagnetic superconductors" also in the title.

  2. I think that there is a typo in the second affiliation.

  3. In the previous version, I expressed my reservation concerning using the word "novel" in the text. In the recent version, one finds the following sentence in the abstract: "Thus our solution describes a novel topological structure- namely a Skyrmion-vortex composite." The skyrmion-vortex excitation has been already proposed. Therefore, the authors should also rectify the above sentence. What I would consider novel, is that the authors discussed such a composite excitations in bulk ferromagnetic superconductors.

  4. In the abstract the authors write: "We also discuss the nature of spin wave dispersion in the ω ∼ \tilde{m} regime which shows a similar pattern in the dispersion curve. " The parameter \tilde{m} is not defined anywhere. Therefore, I suggest the authors define it already in the abstract or provide an alternative form for the above expression.

  5. I think it is useful to mention the work by Martin and Morpurgo (Phys. Rev. B 85, 144505 (2012)) as the first to discuss the possible emergence of Majorana zero modes in ferromagnetic superconductors. This can help improve the motivation of this work since they discuss similar systems. In fact, the proposal of the authors has the advantage that such composite excitations can potentially harbor Majorana zero modes which are localized, in contrast to the flat bands discussed by Martin and Morpurgo, and this can be more advantageous for quantum computing applications.

  6. In page 4, the authors introduce the skyrmion polarity, after the definition of reference 34. However, in the latter work, the authors discussed that the polarity can take two values, i.e., +1 and -1. Here, the authors include also the zero value. But that would imply that the magnetic profile is not a skyrmion. I think it would be best to exclude the zero value for clarity.

  7. In page 4 one finds the sentence "fixed the amplitude M0 of the magnetization". I think this contradicts that later on the amplitude of the magnetization is \mu. The authors can keep the latter symbol throughout.

  8. The photon mass is introduced after equation 13. However, it is not clearly spelled out but instead it appears in a relation. It is better to clearly define it.

  9. Between equations 23 and 26, the authors mention that a potential term is required to be introduced to stabilize magnetic skyrmions in the usual case. I think that the way that this discussion is inserted in the text breaks the flow. The authors can maybe transfer this part as a brief appendix that they can refer to when discussing that in the present case the field is sufficient to stabilize magnetic skyrmions in contrast to the more standard case.

  10. Below equation 25, the authors write "Ginzberg-Landau". I think it is better to write "Ginzburg-Landau" as it appears elsewhere in the text.

  11. It would help the reader if the authors explicitly mention which equation they use to obtain equation 31.

  12. Below equation 31, the authors write "As the spin configuration is assumed to be independent of the z direction,". I think it is better to replace "z direction" by "z coordinate".

  13. It would be helpful for the reader if the authors explicitly define the integrals of equation 33.

  14. An important issue that has to be resolved concerns the result in equation 40. The vorticity N is an integer and cannot be given by the result of 40, unless the r.h.s. is also an integer. Similar to the Little-Parks effect, the vorticity takes the integer value which is the closest to the r.h.s. of equation 40. The authors, instead of differentiating equation 36, should simply "complete the square".

  15. Throughout this work, the use of \hat{n} or \bold{n} for the unit vector is not consistent. The authors should make their notation uniform.

  16. In the discuss between equations 48-50, the authors mention that they employ the Green's function approach to find the magnetic field. This is of course correct but, personally, I find that this discussion unnecessarily complicates the presentation. The authors could simply say that they employ a Fourier transform and obtain the magnetic field.

  17. Equation 58 is solved by keeping terms which are quadratic in frequency in the r.h.s.. Why not doing the same for equation 54?

  18. Above equation 60, one finds the sentence: "Abrikosov vortex along the Z axis". I guess "Z axis" should be "z axis".

  19. In page 17 one finds the phrase: " The small width of the sample". I think it is better to write " The small thickness of the sample".

Recommendation

Ask for minor revision

  • validity: good
  • significance: good
  • originality: good
  • clarity: good
  • formatting: good
  • grammar: good

Author:  Shantonu Mukherjee  on 2025-06-18  [id 5578]

(in reply to Report 1 on 2025-05-31)
Category:
answer to question
reply to objection

Dear SciPost Team,

Many thanks for your email.

We thank the anonymous referee for a very thorough and thoughtful review. We have enclosed our response to the referees comments in the attached pdf file. We hope that the answers provided in the document attached and the changes made in the revised manuscript will be satisfactory to the referee and the manuscript will be accepted for publication without further delay. We shall look forward to your response.

Yours sincerely,

Shantonu Mukherjee, Amitabha Lahiri

Attachment:

Review_Reply.pdf

---

## Round 3 · Author Response

Dear Editor,
We would like to thank SciPost Physics authority for considering our manuscript for review. We also thank the anonymous referee for a very thorough and thoughtful review. We have considered each points raised by the anonymous referee carefully and modified our previous version of the manuscript accordingly. With this letter we are enclosing the revised manuscript. We hope this revised manuscript will convince the concerned authority and it will be considered for publication with no further delay.

Yours Sincerely
Shantonu Mukherjee
Amitabha Lahiri

---

## Round 3 · List of Changes

Details of the changes are given in the reply to reviewers comments. Please have a look.

---

## Round 4 · Referee Report · Anonymous (Referee 2) · 2025-6-19

Weaknesses

None.

Report

I think that the final form of the manuscript is very clear and meets the criteria of the journal. I have also considered the other reports and the authors' response and changes in the manuscript. Overall, I recommend publication in the current form.

Recommendation

Publish (meets expectations and criteria for this Journal)

---

## Round 4 · Referee Report · Anonymous (Referee 1) · 2025-6-19

Strengths

  1. The authors discuss a timely topic, since there has been interest recently on such composite excitations due to their potential role in creating Majorana excitations and implementing quantum computing.

  2. The authors investigate an interesting problem concerning superconducting electrons which are under the influence of the exchange field generated by localized moments. Such a study may find experimental relevance for rare-earth superconductors and reveals how magnetic skyrmions can be engineered in superconducting vortices.

  3. One other novelty of the authors’ approach is that they explored the magnon dispersion in ferromagnetic superconductors with Dzyaloshinskii-Moriya interaction.

  4. The technical approach appears correct.

  5. The results are reasonable and interesting.

  6. Overall this manuscript reads well.

  7. This work can inspire further studies in this field and motivate revisiting the possibility of Majorana zero modes in rare-earth superconductors. Some early ideas when it comes to Majorana modes in such systems were first discussed by Martin and Morpurgo in a PRB of 2012. There, the Majorana zero modes formed flat bands. Here, instead, the authors propose a route to realize localized Majorana modes, which can be advantageous for quantum computing applications.

  8. This version is substantially improved compared to the previous one.

Weaknesses

The authors have rectified all outstanding issues. In fact, they made further improvements to the manuscript that have rendered this work more clear and accessible. I find no weaknesses.

Report

I am in the happy position to recommend this work for publication in its present form. According to my opinion, the authors have put strong efforts to improve their work . Even more, their motivation and target systems have been clarified. I think this is an interesting and timely work that will inspire further activity in the field.

Requested changes

No changes are requested.

Recommendation

Publish (meets expectations and criteria for this Journal)

---

## Editorial Decision

published